# *Schistosoma haematobium* infection is associated with oncogenic gene expression in Cervical Mucosa, with enhanced effects following treatment: A pilot study

Anna M. Mertelsmann[1,2,3,4☯*], Jane K. Maganga[5☯], Myung Hee Lee[4], Maureen Ward[4], Adrian Y. Tan[6], Sheridan F. Bowers[4], Loyce Mhango[5], Danielle de Jong[7], Paul L.A.M. Corstjens[7], Govert J. van Dam[8], Saidi H. Kapiga[5,9], Kathryn M. Dupnik[3,4] Humphrey D. Mazigo[10], Jennifer A. Downs[4,5,11☯], John M. Changalucha[5☯]

**1** Department of Infectious Diseases and Hospital Epidemiology, University Hospital Zurich, Zurich, Switzerland, **2** University of Zurich, Zurich, Switzerland, **3** Division of Infectious Diseases, Department of Medicine, Weill Cornell Medicine, New York, New York, United States of America, **4** Center for Global Health, Weill Cornell Medicine, New York, New York, United States of America, **5** Mwanza Intervention Trials Unit/National Institute for Medical Research, Mwanza, Tanzania, **6** Genomics Resources Core Facility, Weill Cornell Medicine, New York, New York, United States of America, **7** Department of Cell and Chemical Biology, Leiden University Medical Center, Leiden, the Netherlands, **8** Department of Parasitology, Leiden University Medical Center, Leiden, the Netherlands, **9** Department of Infectious Disease Epidemiology and International Health, London School of Hygiene and Tropical Medicine, London, United Kingdom, **10** Department of Parasitology and Entomology, Catholic University of Health and Allied Sciences, Mwanza, Tanzania, **11** Department of Medicine, Weill Bugando School of Medicine, Mwanza, Tanzania

☯ These authors contributed equally to this work.
* annamertelsmann@gmail.com

## Abstract

### Background

*Schistosoma haematobium* is a parasitic worm that infects over 110 million people worldwide, laying eggs that migrate into host urinary and reproductive tracts. While *S. haematobium* is a known carcinogen in the urinary bladder, its role in cervical cancer remains unclear and molecular effects of parasite eggs in genital tissue are largely unknown. Our objective was to characterize cervical transcriptional profiles in women with or without active *S. haematobium* infection and after anti-schistosome treatment.

### Methods

We collected cervical cytobrush samples from women living in areas of Tanzania endemic for *S. haematobium,* before and 4–12 months after praziquantel treatment, and extracted RNA for transcriptome analysis. mRNA was isolated using poly(A) selection and sequencing was performed on an Illumina Hi-Seq4000 platform. Transcript alignment to the human hg19 reference genome and counting were accomplished using the HTSeq package. Genes were assessed for differential expression

**Data availability statement:** Our approved and signed NIH Genomic Data Sharing plan permits sharing of gene-level summary data (gene names and copy numbers), but not raw RNA-Seq reads. Although the consent supports broader data sharing, we are currently bound by the limitations of our GDS plan. Depositing the reads would require us to revise our GDS plan and obtain new written NIH approval for a change in the NIH grant awarded. Instead, we provide all data needed for analysis is in compliance with our approved plan with the current submission as supplementary materials (S5 Table). To ensure the highest level of participant anonymity within this small community, we have removed additional variables that, when combined, could potentially allow for individual identification.

**Funding:** This work was supported by the National Institute for Allergy and Infectious Diseases (R01 AI 168306 and K24 AI 182638 to J.A.D.), and by research training grants to A.M.M. (5T32 AI 007613 and Burroughs Wellcome Fund BWF 102004) and to J.K.M (Fogarty International Center D43 TW 011826). The funders had no role in study design, data collection and analysis, decision to publish, or preparation of the manuscript.

**Competing interests:** The authors have declared that no competing interests exist.

using DESeq2 and Limma. Ingenuity Pathway Analysis (IPA) was employed to identify gene networks altered in the presence of *S. haematobium* infection and following parasitological elimination of infection.

## Results

As part of this pilot study, we enrolled 20 participants with and 19 without *S. haematobium* infection. After adjusting for multiple comparisons, we identified 9 differentially expressed genes in women with versus without infection at baseline, 23 in women with parasitological clearance of infection post-treatment versus with infection at baseline, and 29 in those with parasitological elimination of infection versus without infection at baseline. Most differentially expressed genes were associated with heightened oncogenesis in both women with infection and in those with parasitological clearance of infection after treatment. Using IPA, we identified cancer-related networks and pathways in women with parasitological clearance compared to women with and without infection, as well as pathways involving inflammation and compromised epithelial integrity.

## Conclusion

Women with *S. haematobium* infection and those with recent parasitological clearance were found to have cervical gene alterations that have been reported in various cancers. Our findings suggest a possible increase in cervical cancer risk and susceptibility to secondary infections shortly after treatment. Further research is necessary to ascertain whether altered gene expression after parasitological clearance of *S. haematobium* resolves over time.

---

## Author summary

*Schistosoma haematobium* is a parasitic worm that infects millions globally, mostly in Africa. While its association with bladder cancer is well-documented, its role in cervical cancer is debated. Data on cervical mucosal gene expression in women with this infection are limited. We analyzed cervical transcriptional profiles in women with and without active *S. haematobium* infection, and longitudinally assessed the impact of anti-schistosome praziquantel treatment. Women from Tanzanian villages where *S. haematobium* is highly prevalent were enrolled. RNA was extracted from cervical cytobrush samples, followed by mRNA isolation and sequencing. Gene expression was analyzed using DESeq2, Limma, and pathway analysis tools. Among our cohort of women, differential expression analysis revealed 9 genes in women with versus without *S. haematobium* infection, 23 in women who had parasitological clearance of infection versus women with *S. haematobium* infection, and 29 in women who had parasitological clearance of infection versus women without *S. haematobium* infection. Most genes had

roles in oncogenesis. IPA identified networks and pathways related to cancer and inflammation in women with parasitological clearance compared to women with and without *S. haematobium* infection. In women with *S. haematobium* infection, alterations were observed in cervical mucosal gene expression that have been previously associated with oncogenesis. Gene expression was even more profoundly altered in women who had been recently treated and experienced parasitological clearance.

## Introduction

*Schistosoma haematobium* is a parasite that infects over 110 million people worldwide [1–4]. Adult worms reside in the venules of the genitourinary tract, releasing eggs that can become trapped in tissues. In women, *S. haematobium* eggs in the genital tract mucosa incite cellular immune responses [5,6], leading to local inflammation and fibrosis in the affected tissues. This egg deposition in the genital tract, termed female genital schistosomiasis (FGS), is associated with pain, bleeding, infertility, and likely increases the risk of transmucosal viral infections such as HIV and human papilloma virus (HPV) [7–10]. Praziquantel remains the only treatment for *S. haematobium* infection, and genital mucosal lesions in FGS persist despite praziquantel treatment in many women [11–14]. Very little is known about genital mucosal changes at the molecular level during *S. haematobium* infection. Identification of molecular alterations in the genital tract may uncover novel molecular therapeutic targets for treatment of FGS.

Our current understanding of molecular changes during *S. haematobium* infection in urogenital tissue is derived almost entirely from *S. haematobium*-associated squamous cell carcinoma in the bladder [15,16]. In *S. haematobium*-associated bladder cancer, *S. haematobium* eggs are classified as a Group 1 carcinogen [2,17] and are linked to lower expression of tumor suppressor genes p53 [18,19] and p63 [20], of the pro-inflammatory mediator Cyclooxygenase-2 (COX-2) [21,22], and of inducible nitric oxide synthetase [21,23] in humans. The possibility of similar mucosal molecular effects in the cervix of women with *S. haematobium* infection, which could then increase the risk of cervical cancer, is not well established. In a small cross-sectional study of gene expression in the cervix, women with *S. haematobium* infection had alterations in gene expression related to matrix metalloproteinases, which could damage epithelial tissue integrity, and to cancer signaling [24]. Our understanding of both the short- and long-term effects of *S. haematobium* infection and its treatment with praziquantel remains limited. While praziquantel is commonly used to treat schistosomiasis, there is a significant gap in knowledge regarding its impact on the genital mucosa, especially in the context of sustained or repeated treatments. Insights into potential changes to the mucosal environment, including alterations to tissue structure, immune response, and susceptibility to reinfection and cancer, could guide efforts to treat genital tract pathology in women with FGS.

In this prospective pilot study, we conducted a longitudinal analysis of gene expression in cervical mucosal brushings from women with and without *S. haematobium* infection in Tanzania. By comparing the transcriptomic profiles before and after praziquantel treatment, we aimed to identify potential biomarkers and therapeutic targets for treating FGS, and potentially its associated complications, in women suffering from this neglected tropical disease.

## Methods

### Ethics statement

All women enrolled in the study were adults, and all treatments and procedures were provided at no cost. Ethical approval for the study was granted by the National Health Research Ethics Committee of the National Institute for Medical Research (NIMR/HQ/R.8a/Vol.IX/2446, Dar es Salaam, Tanzania), Weill Cornell Medicine (1612017800, New York, USA), and the London School of Hygiene and Tropical Medicine (27972, London, U.K.).

Additionally, permission was sought from the Simiyu Regional Administrative Secretary (RAS), and from district and village authorities to conduct the study in the region. Written informed consent was provided by all study participants.

## Study sites and population

The study was conducted in rural villages of Itilima district in northern Tanzania where *S. haematobium* is endemic and the prevalence of *S. mansoni* is < 2% [25,26]. As part of the study, women of reproductive age (18 – 50 years) received free screening for schistosome infection, soil-transmitted helminth infections, HIV, sexually transmitted infections (STIs), pregnancy, and cervical cancer. Those eligible for cohort participation were enrolled with a planned follow up of 12 months.

## Population screening

We invited women to be screened for study eligibility with urine, stool, and blood testing, as previously described in detail [24,27]. Participants provided written informed consent prior to any study procedures. Women received individual voluntary counseling and testing for HIV in accordance with the Tanzanian national HIV testing algorithm. Initially, they were tested with a rapid HIV 1/2 test (SD Bioline HIV 1/2 3.0, Abbott Diagnostics, Inc, Korea) followed by a confirmatory test with a separate rapid test if positive (Unigold Recombigen HIV test, Trinity Biotech, Bray County, Wicklow, Ireland). Women confirmed to have HIV infection were referred for free care and treatment at the local health facility. Pregnancy was tested using a point of care beta HCG test (Zhejiang Orient Gene Biotec Co., Ltd Huzhou, Zhejiang, China) at each visit.

Blood specimens were collected by phlebotomy for quantification of serum circulating anodic antigen (CAA) at the National Institute for Medical Research laboratory in Mwanza, with CAA ≥ 30 pg/mL considered positive [28]. Urine and stool samples were collected for *S. haematobium* and *S. mansoni* egg microscopy, respectively. A 10 ml urine sample was filtered and examined under a microscope for *S. haematobium* eggs. The Kato Katz technique was utilized to prepare 5 slides per stool sample to assess for intestinal helminths including *S. mansoni* eggs [29], and any participant with *S. mansoni* infection was excluded. A standard dose of praziquantel (40 mg/kg) was given to any participant diagnosed with schistosome infection by either eggs or CAA at any visit.

## Cohort formation and follow up visits

Eligible women were those with: (1) confirmed *S. haematobium* infection by a positive serum CAA test plus *S. haematobium* eggs visualized on urine microscopy, or (2) women without schistosome ova detected in urine or stool but who had CAA ≥ 30 pg/mL, or (3) *S. haematobium* uninfected as defined by serum CAA < 30 pg/mL and no ova detected in urine or stool. Women with prior diagnosed *S. haematobium* infection who received praziquantel treatment were categorized as having parasitological clearance post-treatment if they had a reduction of the serum CAA < 30 pg/mL, no *S. haematobium* eggs visualized on urine microscopy, and no *S. mansoni* eggs detected by Kato Katz stool preparation at their follow up visits. Women were excluded from the cohort study if they had HIV infection due to possible influence of HIV on the local cervical gene expression. Pregnant women were also excluded from the cohort due to the requirement for endocervical brushing. Any woman who was excluded from the study still received praziquantel treatment if indicated based on serum, urine, and stool results.

Eligible women who agreed to participate in the cohort underwent a private interview by a study nurse in the local language to obtain demographic and clinical information and had a gynecologic examination performed between days 9 and 16 of their menstrual cycle to control for gene expression variations during the menstrual cycle [30]. The study team was trained in FGS lesion recognition and evaluated lesions in comparison to the World Health Organization FGS atlas as a reference during examinations. Findings were confirmed by at least two study team members to maintain diagnostic reliability. During these examinations, an endocervical cytobrush was collected and placed immediately into RNAlater (Thermo Fisher Scientific, Waltham, Massachusetts, USA) for RNA extraction and sequencing. An endocervical swab for *Neisseria gonorrhoeae* and *Chlamydia trachomatis* PCR testing and a high vaginal swab for rapid *Trichomonas vaginalis* testing were collected (see below). Women who were found to have a STI were treated according to the Tanzanian

guidelines for STI management, with empiric antibiotic treatment for cases of cervicitis or abnormal discharge and confirmation of specific indicated antibiotic treatment when PCR results were positive. Sexual partners were also given treatment free of charge. Cervical cancer screening using acetic acid as per the Tanzanian national guidelines was performed at enrollment and month 12. Women with lesions suspicious for cervical cancer were referred and given transport funds to reach the Nkoma Health Center for further management, provided free of charge by the Tanzanian national health plan. Team members were not blinded to *S. haematobium* infection status.

Participants were invited to return for follow up visits for up to 12 months after enrollment. At each follow-up, serum was collected for schistosome CAA testing and urine was collected for pregnancy testing and microscopy. Women who had schistosome infections received observed treatment with praziquantel (40 mg/kg) each time they were found to have a serum CAA ≥ 30 pg/mL or to have schistosome eggs in urine. Women who were found to be pregnant received praziquantel according to World Health Organization guidelines and were excluded from further follow-up. Testing for *S. mansoni* in stool was repeated at 6 and 12 months and those found to be positive were treated and excluded from further follow-up.

## Sample processing and testing I

DNA was extracted from endocervical swabs for *N. gonorrhoeae* and *C. trachomatis* quantitative polymerase chain reaction (qPCR) testing (Artus CT/NG QS-RGQ, Qiagen, Hilden, Germany) at the laboratory in Mwanza. Vaginal swabs were tested on site at the time of the gynecological examination for *T. vaginalis* using a rapid point-of-care test (OSOM, Sekisui Diagnostics, CA, USA). Endocervical cytobrushes preserved in RNAlater were stored at -80 ºC at the NIMR laboratory and shipped to New York on dry ice with temperature monitoring for further analysis. At baseline visits, blood samples were collected and centrifuged to isolate serum, which was then stored at −80°C. The serum samples were subsequently tested for antibodies against HSV-2 using a commercially available ELISA kit (Kalon HSV-2 IgG ELISA, Kalon Biological Ltd, UK), according to manufacturer's instructions.

## RNA extraction and purification

RNA was extracted from cervical cytobrush samples as previously described [24]. In brief, cytobrushes were disrupted using a Bead Mill 4 bead beater (Thermo-Fisher Scientific) in nuclease-free tubes containing metal beads (Thermo-Fisher Scientific) for 1 minute at medium speed (3 m/second or 90 × g). Afterwards the RNAlater was removed and centrifuged for 6 minutes at 15 000 rpm. The supernatant was removed and added back to the tubes with metal beads. After bead beating at a speed of 4 m/second (120 × g) for an additional minute, removal of RNAlater and centrifugation was repeated. RNA was extracted from the collected supernatant with the RNeasy Mini Kit with on-column DNase digestion, according to product instructions (Qiagen). RNA integrity was determined with a Bioanalyzer 2100 (Agilent Technologies, Santa Clara, CA), and the RNA concentration was measured using the NanoDrop 8000 (ThermoFisher Scientific).

Samples with RNA integrity numbers >2 were submitted for next-generation RNA sequencing (RNA-Seq). RNA sample preparation and RNA-Seq were performed in 2 batches. Follow up specimens were sequenced in the same batch as baseline samples from the same participant.

## RNA library preparation and RNA-Seq

RNA sample library preparation and RNA-Seq were performed by the Weill Cornell Genomics Core laboratory at Weill Cornell Medicine. In brief, messenger RNA was prepared using the NEB Ultra II Directional RNA Library Preparation kit (New England Biolabs, Ipswich, MA), according to the manufacturer's instructions. Normalized cDNA libraries were pooled and sequenced on Illumina NovaSeq 6000 sequencer with paired-end 50-bp reads for 100 cycles. Raw sequencing reads in BCL format were processed through bcl2fastq 2.19 (Illumina) for FASTQ conversion and demultiplexing.

After trimming the adaptors with cutadapt (version1.18, https://cutadapt.readthedocs.io/en/v1.18/), RNA reads were aligned and mapped to the GRCh38 human reference genome by STAR (Version2.5.2) (https://github.com/alexdobin/STAR) [31], and transcriptome reconstruction was performed by Cufflinks (Version 2.1.1) (http://cole-trapnell-lab.github.io/cufflinks/).

## Statistics

Demographic and clinical characteristics were summarized as numerical values with percentages or as median values with interquartile ranges. Statistical comparisons were conducted using Fisher's Exact test for categorical variables and the Wilcoxon rank sum test for continuous variables. Principal component analysis was employed to visualize the underlying sources of variation in the dataset.

We determined that we would compare cervical mucosal gene expression between women in three different groups: baseline women without *S. haematobium* infection, baseline women with *S. haematobium* infection, and women with parasitological clearance post-treatment who had no *S. haematobium* eggs detected in urine and had serum CAA < 30 pg/mL at follow-up. We performed pre-filing to remove genes with very low counts and retained only genes with a count of at least 5 in the majority of sequenced samples for the subsequent differential expression analysis. To make these comparisons, gene expression from cervical cytobrush samples were analyzed using DESeq2 and Limma in R (version 4.2.3) [32].

For two-group comparisons, genes with at least 2-fold changes were sorted by p-values in an increasing order and rank of the gene was assigned, once by DESeq and once by Limma. Based on the sum of the two ranks, the top nine genes that were robustly ranked on the top of the list were selected based on the average rank and box plots were generated. Corrected P values were calculated using the Benjamini-Hochberg method adjustment for multiple testing to control the false discovery rate at ≤0.05.

All genes identified as being significantly differentially expressed with a $p_{adj}$ of ≤0.05 from the DESeq analysis were submitted for IPA to identify altered pathways (Qiagen) and for Webgestalt Kyto Encyclopedia of Genes and Genomes (KEGG) pathways analysis using overrepresentation enrichment analysis [33].

## Terminology

Differentially expressed refers to genes that exhibit statistically significant differences in copy numbers of genes expressed between the experimental and control conditions. These differences are determined by applying thresholds for fold change (e.g., > 2 or ≤2) and statistical significance (e.g., adjusted p-value <0.05).

Using IPA, genes can be predicted to be activated, as part of molecular pathways, biological processes, or upstream regulators that are inferred by IPA to have increased activity based on the pattern of input gene expression data. This prediction is made using the z-score algorithm, where a positive z-score indicates activation. Genes are predicted to be inhibited when they are part of molecular pathways, biological processes, or upstream regulators that are inferred by IPA to have decreased activity based on the pattern of input gene expression data. A negative z-score indicates inhibition. Canonical refers to well-established, curated signaling or metabolic pathways that are broadly recognized in the scientific literature. IPA uses these canonical pathways to provide a framework for interpreting gene expression data and identifying pathway-level changes.

## Software

Genes with the lowest P values for differences that had at least 2-fold changes were used to create box plots in R (version 4.2.3). Plots of CAA values over time were created with GraphPad Prism (La Jolla, USA). Figures were created with BioRender (Toronto, Canada).

## Results

### Cohort characteristics and demographics

Women were screened for study eligibility and enrolled into the cohort between June 2021 and August 2022. A total of 63 women without HIV had baseline cervical samples available for RNA extraction (Fig 1).

Thirty women had confirmed *S. haematobium* infection. Among them, seven had positive results for serum CAA plus *S. haematobium* eggs visualized on urine microscopy. Twenty-three women had positive results for CAA but no schistosome eggs visualized in urine or stool specimens [34]. The other 33 were negative for schistosome infection by CAA and microscopy of urine and stool.

After RNA extraction, 24 did not meet the criteria for proceeding to RNA-Seq, due to either DNA contamination or to insufficient RNA quality or quantity. Despite multiple attempts to remove DNA with DNase treatment, not enough RNA could be obtained from the DNA contaminated samples to be sufficient for sequencing. Excluded women did not have any significant clinical or demographic differences in comparison to the included women.

Ultimately, a total of 39 women had samples which yielded sufficient quality and quantity RNA to undergo RNA sequencing. The baseline characteristics of the included women are shown in Table 1.

Of the included women, 20 (20/39 [50%]) were diagnosed with *S. haematobium* infection with a median serum CAA of 421.8 [IQR, 159.8-2992.3] pg/ml at baseline. Nine of these had eggs seen on urine microcopy with a median egg count of 6 [3–7] eggs/10ml and a median CAA of 1827 [230.9-3905.4] pg/mL. Among the 11 who had no eggs in urine, the median CAA was 368 [86.7-1033.1] pg/mL.

Women without *S. haematobium* infection had a trend towards their children more frequently having multiple fathers compared to women with *S. haematobium* infection (17/19 (89.5%) versus 10/17 (58.8%), p = 0.055), although this did not reach statistical significance. No other differences were observed in demographic or behavioral characteristics, nor in the prevalence of gonorrhea, chlamydia, or trichomoniasis, nor of HSV-2 seropositivity, between those with and without schistosome infection. No intestinal helminths other than hookworm infection were identified by Kato-Katz method.

Of the 20 women with *S. haematobium* infection, 13 women had samples available for RNA sequencing at follow-up after praziquantel treatment at months 4, 6, and/or 12. Two samples collected from these 13 women at baseline were excluded due to DNA contamination but follow up samples were available. A total of 19 follow up samples after praziquantel treatment from these 13 women with *S. haematobium* infection were able to be sequenced and included in the analysis. Of these, 13 samples were collected when women had parasitological clearance of infection (CAA < 30 pg/ml), while 6 samples were collected when women were still infected despite praziquantel treatment, for which they received another dose of praziquantel. Fig 2 displays the trend in serum CAA of these 13 women over time.

At follow-up, a total of five STIs were diagnosed. *Trichomonas* infections accounted for three of these cases, with two women diagnosed at their 6-month follow-up visits and one woman diagnosed at both her 6-month and 12-month visits. Additionally, two cases of gonorrhea were identified, with one diagnosed at the 4-month visit and the other at the 6-month visit.

Due to the low number of samples, a paired analysis could not be performed for women with persistent infections and comparisons between different months post-treatment could not be made. We therefore performed three comparisons, each of which is described in detail in the following sections: (1) gene expression in women with versus without *S. haematobium* infection at baseline; (2) gene expression in women with parasitological clearance post-praziquantel treatment versus women with baseline *S. haematobium* infection; and (3) gene expression in women with parasitological clearance post-praziquantel treatment versus women without *S. haematobium* infection at baseline (Fig 3).

There was no difference in prevalence of co-infections, including gonorrhea, chlamydia, trichomoniasis, or HSV-2, between those who did and did not clear *S. haematobium* infection at their follow up visits.

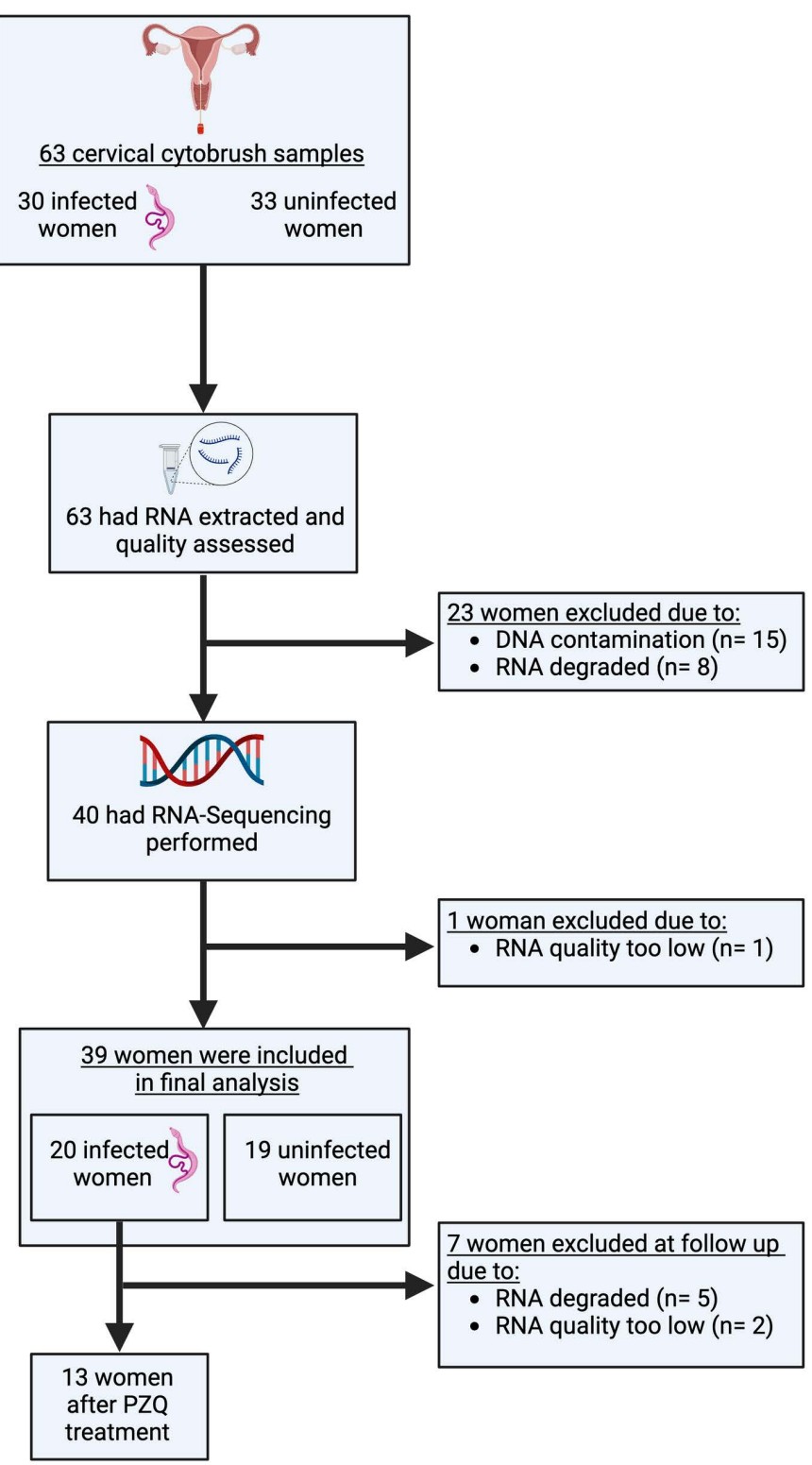

**Fig 1. Flow chart of sample collection, processing, and RNA-Seq performance.** A total of 63 women had cervical cytobrush samples available for testing. RNA was extracted from all available samples. Out of the 63 women, 24 women had samples which could not be sequenced and were excluded. Reasons for exclusion were DNA contamination (n = 15), degraded RNA (n = 8) and low-quality RNA (n = 1). Created in BioRender. BioRender.com/t06f390.

**Table 1. Characteristics of study participants with and without *Schistosoma haematobium* infection at baseline.**

| Factors | Women with *S. haematobium* Infection (n = 20) Number (%) or Median [IQR] | Women without *S. haematobium* Infection (n = 19) Number (%) or Median [IQR] |
|---|---|---|
| **Demographic characteristics** | | |
| Age (years) | 30 [22.5-41] | 36 [23-46] |
| Years of schooling | 7 [7–7] | 7 [0-7] |
| **Marital Status** | | |
| Single | 2 (10%) | 1 (5.3%) |
| Married | 17 (85%) | 17 (89.5%) |
| Divorced | 1 (5%) | 0 (0%) |
| Widowed | 0 (0%) | 1 (5.3%) |
| Ever received past praziquantel treatment* | 7 (36.8%) | 10 (52.6%) |
| **Reproductive and sexual health*** | | |
| Age at first pregnancy (years) | 18 [16–19] | 19 [17–20] |
| Number of pregnancies | 6 [3–8] | 5 [2–9] |
| Number of children born | 5 [2–7] | 5 [2–6] |
| Children have multiple fathers | 10 (58.8%) | 17 (89.5%) |
| Reported infertility | 1 (5.3%) | 4 (21.1%) |
| Age at first sex (years) | 17 [15 –18] | 18 [15 –19] |
| Had greater than one sexual partner in past year | 2 (11.1%) | 3 (17.7%) |
| Accepted money for sex (past year) | 3 (15.8%) | 4 (21.1%) |
| Ever used contraceptives[a] | 7 (38.9%) | 10 (52.6%) |
| Contraceptive use (past 2 months) | 3 (16.7%) | 5 (26.3%) |
| Vaginal cleansing[b] (past 3 months) | 17 (89.5%) | 14 (73.7%) |
| Cleansing frequency among those who cleanse (times per week) | 14 [7 –14] | 14 [14 –14] |
| **Reported genital tract symptoms (past year)*** | | |
| Painful intercourse | 10 (52.6%) | 13 (68.4%) |
| Bleeding at intercourse | 2 (10.5%) | 7 (36.8%) |
| Painful ulcer | 4 (21.1%) | 9 (47.4%) |
| Genital warts | 12 (63.2%) | 11 (61.1%) |
| **Current co-infections** | | |
| Hookworm infection | 1 (5%) | 3 (15.8%) |
| *N. gonorrhoeae* infection | 1 (5%) | 1 (5.3%) |
| *C. trachomatis* infection | 1 (5%) | 1 (5.3%) |
| *T. vaginalis* infection | 4 (20%) | 3 (15.8%) |
| Any current genital mucosal STI** | 5 (25%) | 5 (26.3%) |
| Herpes Simplex Virus 2 seropositivity | 14 (70%) | 13 (68.4%) |
| History of past STI treatment | 4 (21.1%) | 3 (15.8%) |
| **Gynecology examination findings** | | |
| Abnormal discharge | 9 (45%) | 10 (52.6%) |
| Contact bleeding | 2 (10%) | 0 (0%) |
| Abnormal blood vessels | 8 (40%) | 6 (31.6%) |
| Ulcer | 6 (30%) | 3 (15.8%) |
| Sandy patches | 5 (25%) | 3 (15.8%) |

*Nonmissing data reported for each characteristic. No characteristic had more than 3 missing data points.

**Some participants were coinfected with *T. vaginalis*, *C. trachomatis*, and/or *N. gonorrhoeae*.

[a]Contraceptive use included hormonal as well as barrier contraceptives.

[b]Vaginal cleansing is a widespread practice in Tanzania. Methods commonly include douching or external washing with water, soap, herbal remedies or antiseptics, typically applied with fingers or cloth. These practices may occur for hygiene purposes, in relation to menstruation, or as part of sexual preparation or aftercare [35].

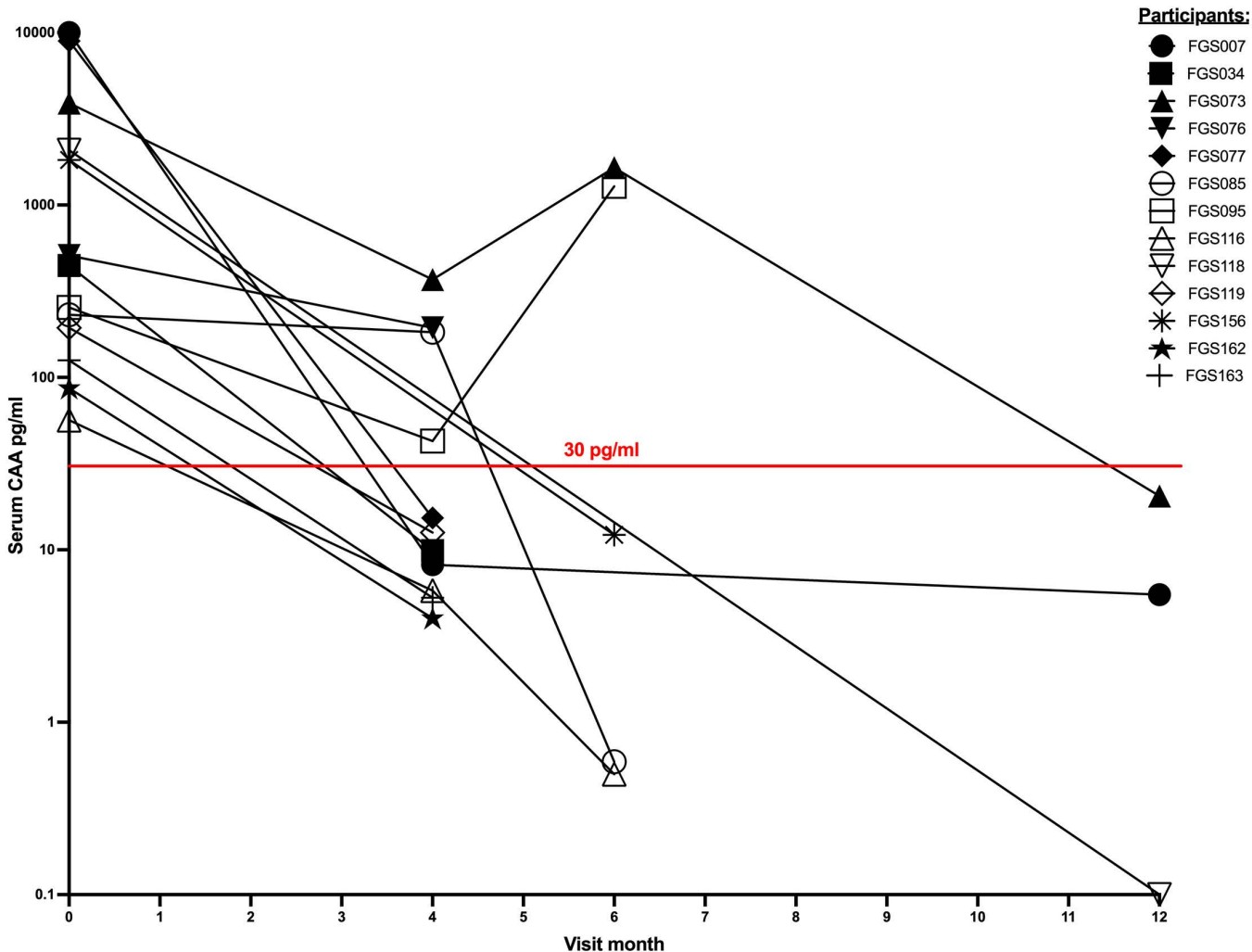

**Fig 2. Serum CAA trend after praziquantel therapy.** Serum CAA trend over time of women at follow up visits. CAA values of samples from months 4, 6, and 12 which were available for analysis are shown. The red line marks the CAA positivity cut-off value of 30 pg/ml. All women were infected at baseline and received observed praziquantel treatment. Participants with CAA values below the red line were considered to have parasitological clearance of *S. haematobium* infection, while participants with CAA values equal to or above 30pg/ml were considered infected and received repeat observed praziquantel treatment. Eleven women had samples from their 4-month follow up visit. Of these, 7 had parasitological clearance and 4 were persistently infected. Five women had samples from their 6-month follow up visit. Of these, 3 had parasitological clearance and 2 were persistently infected. All 3 women who had samples available from their 12-month visit had achieved parasitological clearance. Graph created with GraphPad Prism.

### Gene expression in women with and without S*. haematobium* infection

We had a total of 37 baseline samples available for RNA sequencing. Of these, 18 (49%) were from women with *S. haematobium* infection. Principal component analyses of the overall gene expression did not show major separation between women with *S. haematobium* infection, those without infection, those with parasitological clearance post-praziquantel, or those with persistent infection at 4, 6 and/or 12 months after treatment (see S1A and S1B Fig as S1 Fig). We identified 40 genes that were differentially expressed in the cervix of women with versus without infection using DESeq, 73 using Limma, and 9 that were identified by both. Fig 4 illustrates box plots depicting the transcript levels of these 9 genes comparing women with and without *S. haematobium* infection. The corresponding gene functions and statistics are listed in S1 Table.

### Comparison #1

**Women with *S. haematobium* infection at baseline**

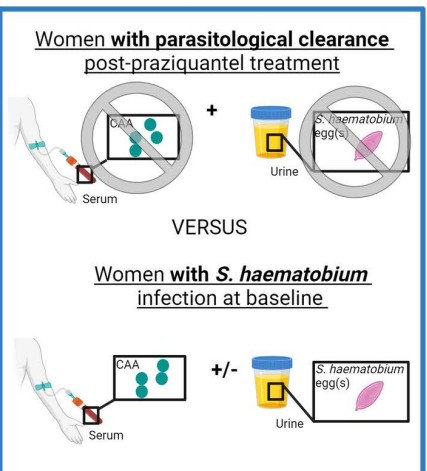

**VERSUS**

**Women without *S. haematobium* infection at baseline**

### Comparison #2

**Women with parasitological clearance post-praziquantel treatment**

**VERSUS**

**Women with *S. haematobium* infection at baseline**

### Comparison #3

**Women with parasitological clearance post-praziquantel treatment**

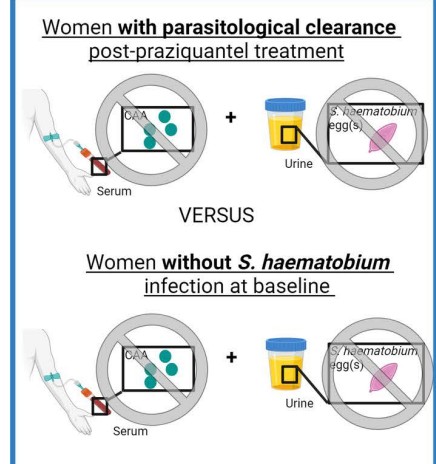

**VERSUS**

**Women without *S. haematobium* infection at baseline**

**Fig 3. Illustrates the three comparisons performed and the testing used to define each group.** Comparison 1 assesses the gene expression in women with versus without S. haematobium infection at baseline; Comparison 2 assesses the gene expression in women with parasitological clearance post-praziquantel treatment versus women with baseline S. haematobium infection; and Comparison 3 assesses gene expression in women with parasitological clearance post-praziquantel treatment versus women without S. haematobium infection at baseline. Created in BioRender, https://BioRender.com/r34l610.

Four of these 9 genes [BLK proto-oncogene, Src family tyrosine kinase (BLK), Long Intergenic Non-Protein Coding RNA 2084 (LINC02084), Trichohyalin (TCHH), and TCL1 family AKT coactivator A (TCL1A)] have been linked to malignancies [36–40] (S1 Table).

There were no significant pathways identified by IPA comparing women with and without *S. haematobium* using a cutoff of $p_{adj} \leq 0.05$. No KEGG pathways showed significant enrichment when using a false-discovery rate cutoff of 0.05.

### Gene expression in women with parasitological clearance post-praziquantel versus women with baseline *S. haematobium* infection

We identified 43 genes that were differentially expressed in the cervix of women with parasitological clearance post-praziquantel treatment versus women with baseline *S. haematobium* infection using DESeq, 144 using Limma, and 23 by both. Fig 5 displays box plots of the top nine genes that were differentially expressed.

Five of these top most differentially expressed genes have been implicated in various malignancies. These include: C-X-C motif chemokine ligand 14 (CXCL14), Long intergenic non-protein coding RNA 592 (LINC00592), Lymphocyte antigen 6 family member K (LY6K), Nuclear receptor subfamily 1 group D member 1 (NR1D1) and NACHT and WD repeat domain containing 2 (NWD2) [41–46]. The comprehensive list of the genes and functions is shown in S2 Table.

When examining women with parasitological clearance post-praziquantel versus women with *S. haematobium* infection, the top canonical pathway by IPA with a false-discovery rate cutoff of 0.05 was Sertoli cell-germ cell junction signaling pathway ($p = 5.15 \times 10^{-7}$). Genes within this pathway are listed in S3 Table, which indicates the top IPA pathway and altered genes in that pathway for each of the three comparison groups.

Other altered canonical pathways included RHO GTPase cycle ($p = 5.61 \times 10^{-7}$), Sertoli cell-Sertoli cell junction signaling ($p = 1.25 \times 10^{-6}$), complement cascade ($p = 1.29 \times 10^{-6}$), CDK5 signaling ($p = 6.32 \times 10^{-5}$), Fc gamma receptor (FCGR) dependent phagocytosis ($p = 6.39 \times 10^{-5}$), cell surface interactions at the vascular wall ($p = 1.03 \times 10^{-4}$), ethanol degradation II ($p = 1.23E \times 10^{-4}$), germ cell-Sertoli cell junction signaling ($p = 1.43 \times 10^{-4}$) and xenobiotic metabolism signaling ($p = 1.49 \times 10^{-4}$).

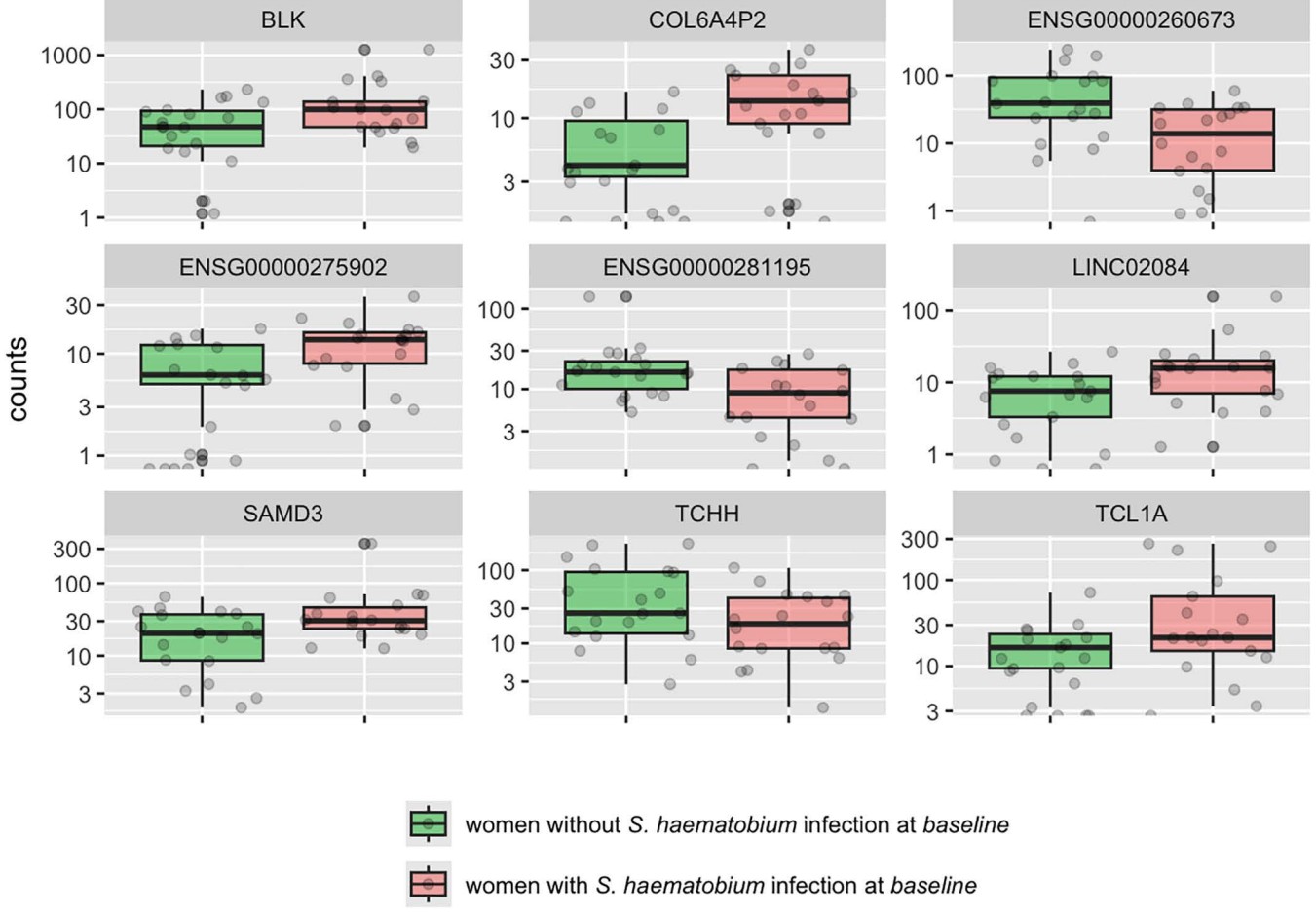

women without *S. haematobium* infection at *baseline*

women with *S. haematobium* infection at *baseline*

**Fig 4. Transcript count in cervical cells of the top 9 differentially expressed genes in women with and without *S. haematobium* infection.** Differences in transcript count in cervical cells for 9 genes in women with and without *S. haematobium* infection at baseline. The plots display median values (dark horizontal bar) and interquartile ranges (boxes), with error bars representing 1.5 times the interquartile range or the minimum/maximum values. BLK encodes BLK proto-oncogene; COL6A4P2 collagen type VI alpha 4 pseudogene 2; ENSG00000260673 is a novel transcript; ENSG00000275902 is a novel transcript; ENSG00000281195 is a novel transcript; LINC02084 Long Intergenic Non-Protein Coding RNA 2084; SAMD3 sterile alpha motif domain containing 3; TCHH trichohyalin; TCL1A TCL1 family AKT coactivator A.

Predominant altered networks identified by IPA, in descending order of statistical significance, included lipid metabolism (small molecule biochemistry, vitamin and mineral metabolism), nucleic acid metabolism, organ injury, cancer, and renal and urological disease.

No significantly enriched KEGG pathways, using a false-discovery rate cutoff of 0.05, were identified.

**Gene expression in women with parasitological clearance post-praziquantel versus women without *S. haematobium* infection at baseline**

We identified 204 genes that were differentially expressed in the cervix of women who had parasitological clearance of infection at follow-up versus women without *S. haematobium* infection at baseline using DESeq, 75 using Limma, and 29 using both. In Fig 6, box plots present the top 9 genes.

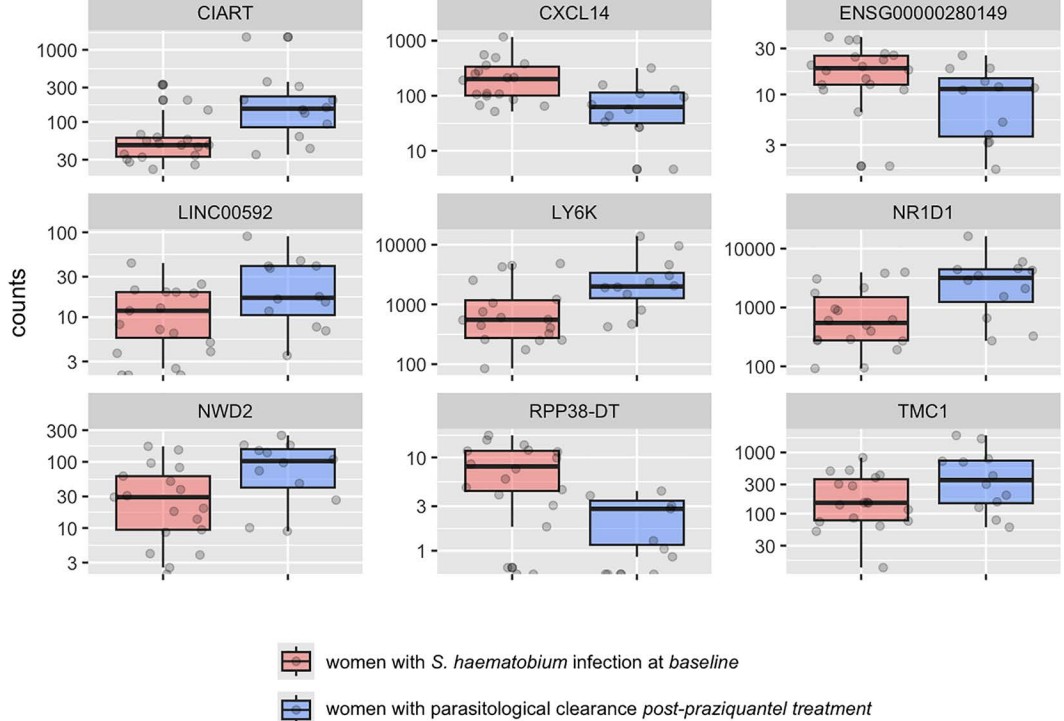

**Fig 5. Transcript count in cervical cells of the top 9 differentially expressed genes in women with parasitological clearance post-praziquantel treatment versus women with baseline *S. haematobium* infection.** Differences in transcript count in cervical cells for the top 9 genes found to be differentially expressed in women with parasitological clearance of *S. haematobium* infection versus women with baseline infection. The plots display median values (dark horizontal bar) and interquartile ranges (boxes), with error bars representing 1.5 times the interquartile range or the minimum/maximum values. CIART encodes circadian associated repressor of transcription; CXCL14 C-X-C motif chemokine ligand 14; ENSG00000280149 is an uncategorized gene; LINC00592 long intergenic non-protein coding RNA 592; LY6K lymphocyte antigen 6 family member K; NR1D1 nuclear receptor subfamily 1 group D member 1; NWD2 NACHT and WD repeat domain containing 2; RPP38-DT RPP38 divergent transcript; TMC1 transmembrane channel like 1.

Genes C-X-C motif chemokine ligand 14 (CXCL14), Interleukin 1 receptor like 1 (IL1RL1), Kinase insert domain receptor (KDR), Nuclear receptor subfamily 1 group D member 1 (NR1D1), Prolyl 3-hydroxylase 2(P3H2), PRDM16 divergent transcript (PRDM16-DT/ LINC00982) and Rap guanine nucleotide exchange factor 5 (RAPGEF5) have all been reported to be associated with malignancies [41,45,47–51]. Genes and functions are listed in S4 Table.

Comparing women with parasitological clearance post-praziquantel and women without *S. haematobium* infection, the top canonical pathway identified by IPA using $p_{adj}$ of ≤0.05 as a cutoff was activin inhibin signaling pathway ($p = 1.13 \times 10^{-6}$). The corresponding genes within this pathway are listed in S3 Table. This was followed by complement cascade ($p = 1.34 \times 10^{-5}$), Fc gamma receptor (FCGR) dependent phagocytosis ($p = 3.99 \times 10^{-5}$), pathogenesis of multiple sclerosis ($p = 4.08 \times 10^{-5}$), pathogen induced cytokine storm signaling pathway ($p = 4.73 \times 10^{-5}$), primary immunodeficiency signaling ($p = 1.28 \times 10^{-4}$), hepatic fibrosis/ hepatic stellate cell activation ($p = 1.75 \times 10^{-4}$), Fc epsilon receptor (FCERI) signaling ($p = 2.63 \times 10^{-4}$), pulmonary fibrosis idiopathic signaling pathway ($p = 3.10 \times 10^{-4}$) and IL-12 signaling and production in macrophages ($p = 5.17 \times 10^{-4}$).The top identified networks by IPA were associated with organismal injury, cellular development, cell morphology, reproductive disease and cancer.

No KEGG pathways were found to be significantly enriched at a false-discovery rate cutoff of 0.05.

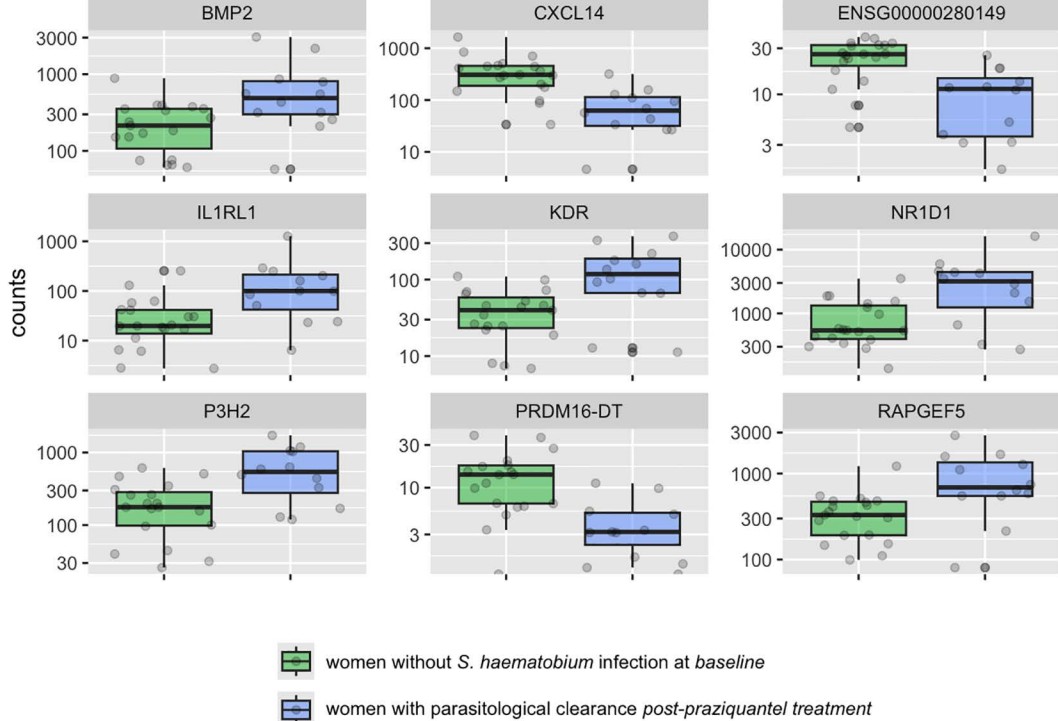

**Fig 6. Transcript count in cervical cells of the top 9 differentially expressed genes in women women with parasitological clearance post-praziquantel versus women without *S. haematobium* infection at baseline.** Differences in transcript count in cervical cells for the top 9 representative genes found to be differentially expressed in women with parasitological clearance after praziquantel treatment and without infection at baseline. The plots display median values (dark horizontal bar) and interquartile ranges (boxes), with error bars representing 1.5 times the interquartile range or the minimum/maximum values. BMP2 encodes bone morphogenetic protein 2; CXCL14 C-X-C motif chemokine ligand 14; ENSG00000280149 is an uncategorized gene; IL1RL1 interleukin 1 receptor like 1; KDR kinase insert domain receptor; NR1D1 nuclear receptor subfamily 1 group D member 1; P3H2 prolyl 3-hydroxylase 2; PRDM16-DT PRDM16 divergent transcript; RAPGEF5 Rap guanine nucleotide exchange factor 5.

Fig 7 displays the graphical summaries of the major biological themes identified by IPA as being altered in women with parasitological clearance post-praziquantel in comparison to women with *S. haematobium* infection at baseline (Fig 7A), and in women with parasitological clearance post-praziquantel compared to women who were uninfected at baseline (Fig 7B).

Key biological themes that were altered in women with parasitological clearance post-praziquantel versus women with baseline *S. haematobium* infection included those associated with oncogenesis-related processes like vasculogenesis, tumor cell activation, and reduced apoptosis of tumor cells (Fig 7A). Additionally, shifts in epithelial cell relationships, including cell junction signaling and extension of cellular protrusions, were noted. In this context, the genes hepatocyte nuclear factor 4 gamma (HNF4G), hepatocyte nuclear factor 4 alpha (HNF4A), geminin DNA replication inhibitor (GMNN), SRY-box transcription factor 1 (SOX1) and 3 (SOX3) were predicted to be inhibited in women with parasitological clearance post-praziquantel compared to women with baseline *S. haematobium* infection.

Similarly, comparing women with parasitological clearance post-praziquantel versus women without *S. haematobium* at baseline, we found an increase in oncogenic pathways related to vasculogenesis, tumor cell migration, cell invasion, and the proliferation of tumor cells. Women with parasitological clearance post-praziquantel, compared to those without infection at baseline, had lower expression of the gene Collagen type I alpha 1 (COL18A1) (Fig 7B).

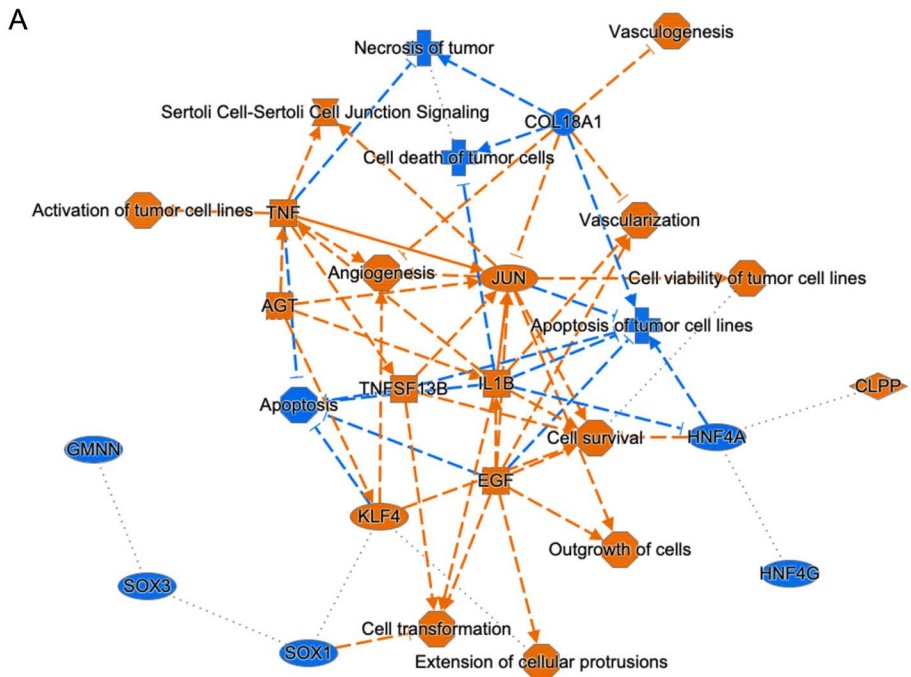

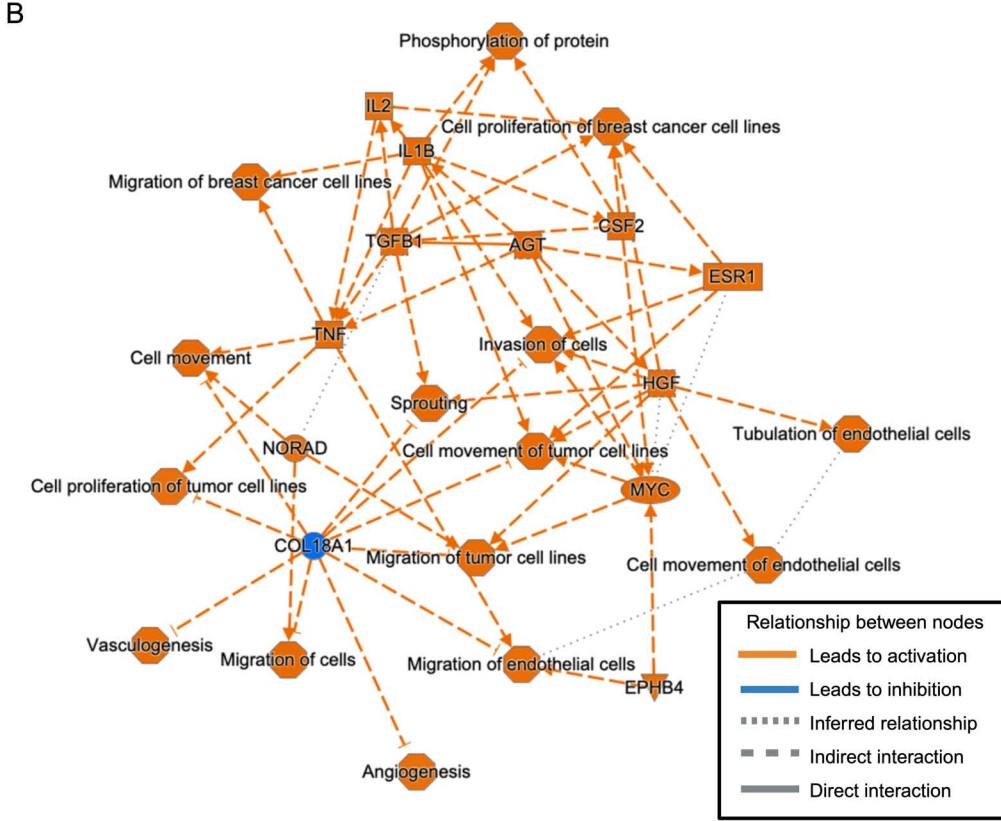

**Fig 7. Graphical summary of identified biological themes identified by Ingenuity Pathway Analysis.** Biological themes and their relations to each other identified by IPA in women with parasitological clearance of *S. haematobium* infection post-praziquantel treatment versus women with baseline *S. haematobium* infection (A) and women with parasitological clearance of infection post-praziquantel treatment versus women without *S. haematobium* infection at baseline (B).

## Discussion

To the best of our knowledge, this pilot study is the first to suggest a link between *S. haematobium* infection and genital malignancy using gene expression in the female genital tract, and then to follow mucosal gene expression longitudinally post-treatment. Our findings build on a previous cross-sectional study conducted by our group in a different region of Tanzania [24]. Other cervical transcriptomic studies thus far have linked altered gene expression to HPV, cervical cancer, or gynecological syndromes like endometriosis but not to parasitic infections [49,52–54]. Here, we describe altered cervical gene expression that resembles expression from bladder mucosa in individuals with *S. haematobium*-associated bladder cancer, including genetic and epigenetic changes related to cell hyperplasia and cancer [17,55,56]. *S. haematobium* eggs are a group I carcinogen in the bladder [55] and it is biologically plausible that eggs in female genital tract could similarly increase risk of genital tract cancer. Further, observed alterations in genital tract expression of immune-related genes could be consistent with heightened risk of HPV acquisition or persistence and development of cervical cancer [7]. These pilot data serve as a guide for the design and conduct of larger confirmatory studies with longer follow-up that are able to define mechanisms by which *S. haematobium* in the genital tract may promote oncogenesis.

We found that women with *S. haematobium* infection compared to uninfected women, and even more that women who experienced parasitological clearance of infection compared to uninfected women, had altered genital tract expression of genes that have been associated with cancers. Importantly, these alterations in host gene expression may indicate a potential path to tumorigenesis in the cervix that could occur independently of HPV. Tissue studies indicate that *S. haematobium* eggs secrete reactive metabolites that lead to formation of catechol estrogens, which are linked to carcinogenesis [57]. In urothelial cells, *S. haematobium* eggs promote cellular proliferation, downregulate the p53 tumor suppressor pathway, and decrease apoptosis [18,58], and our data suggest that eggs may have similar effects in genital mucosa. For example, women with versus without *S. haematobium* infection had upregulation of LINC02084 (Fig 4), a long non-coding RNA associated with proliferation and cancer progression and a risk predictor in several cancers, including head and neck, renal cell and colon cancer [37,59,60]. Further, in women with parasitological clearance, the top identified canonical pathway altered was "Sertoli cell-germ cell junction pathway," which has also been described in women with ovarian tumors [61]. Moreover, in these women, the IPA analysis anticipated inhibition of multiple genes (HNF4G, HNF4A, GMNN, SOX1, SOX3) that have been described to be downregulated in multiple types of cancer [62–67], including cervical cancer [68–71], and upregulation of the transforming growth factor-beta 2 pathway, also found to be altered when urothelial cells were exposed to *S. haematobium* eggs. [18] Together, these data suggest mechanisms that may promote a shift toward an oncogenic state, both during *S. haematobium* infection and possibly even more in the early months after parasitological clearance of infection. As this analysis is based on RNA expression data, we suggest that these observed differences reflect inflammation-driven transcriptional changes rather than direct alterations to the genomic DNA, though our RNA-based data cannot distinguish whether differential expression is due to transient inflammatory signaling or to stable mutational events contributing to oncogenesis.

Additional mucosal alterations in women with parasitological clearance post-treatment, compared to infected women, appear consistent with compromised tissue structure. For example, women with parasitological clearance were predicted by IPA to have inhibition of claudins, cell adhesion molecules, and tight junction proteins (S3 Table), which can impair the epithelial barrier and cytoskeleton integrity and have also been linked to inflammation [72,73]. The death of adult worms leads to the release of parasite-derived antigens, which in turn provoke an immune response [74,75]. This may result in widespread inflammation and tissue damage beyond the vasculature which may initiate pathways of tissue remodeling and fibrosis, potentially mirroring the host response observed around schistosome eggs embedded in tissue. Women with parasitological clearance, compared to infected women, also showed pro-inflammatory gene activation, with upregulation of genes associated with immune activation and epithelial cell transformation (S3 Table). Altered gene expression related to epithelial integrity and mucosal inflammation might be expected during active schistosome infections, when immunogenic parasite eggs migrate through mucosal tissue in order to be excreted by the host and complete the parasite

life cycle. However, the finding that genital epithelial integrity may be even further impaired, and inflammation further enhanced, in the months following parasitological clearance of infection was surprising.

We initially hypothesized that, after treatment, altered gene expression in the genital mucosa would at least partially revert towards the uninfected state. Instead, women with parasitological clearance post-praziquantel treatment, compared both to women with infection and to women without, exhibited severer cervical transcriptomic changes that were associated with further increased oncogenic and inflammatory activity and reduced mucosal integrity. These changes, potentially due to praziquantel-induced killing of adult worms as indicated by a decrease of serum CAA to <30 pg/mL, may revert over time as the host heals. Perhaps analogously, one study that investigated HIV-1 RNA viral loads after effective eradication of *S. mansoni* worms in people living with schistosome-HIV-1 co-infection reported a transient elevation in HIV-1 RNA viral load at 1 month, which normalized after 5 months [76]. A second similar study did not quantify viral loads at 1 month post-praziquantel treatment, but reported that viral loads were lower than baseline after 3 months [77]. Due to the small sample size, women included in the parasitological clearance group in our pilot study had samples collected at a median of 4 months post-treatment and we were not able to examine a differential post-parasitological clearance effect over time.

Importantly, our study did not diagnose FGS directly by identification of parasite eggs or DNA in the genital tract. Rather, we studied women with active *S. haematobium* infection, among whom at least half can be estimated to have genital tract involvement [2,78,79]. The significant differences in genital tract gene expression in women with and without *S. haematobium* infection, and in those who became CAA negative versus those who were infected, supports our decision to classify women by circulating schistosome antigen positivity. It is possible that we would have found even more pronounced gene expression differences if we had restricted analysis to women with confirmed genital *S. haematobium*. Alternatively, it is possible that active *S. haematobium* infection involves the genital tract in the larger majority of women than previously believed, or that systemic parasitic effects are sufficient to evoke genital mucosal changes.

This pilot study has strengths and limitations. We caution that our results should be interpreted as hypothesis-generating rather than conclusive. These changes were modest in magnitude and involve a limited set of genes selected rigorously based on $p < 0.05$ by both Limma and DESeq. Given the small sample size and potential confounding from co-infections, these findings require validation in future studies using complementary techniques such as quantitative PCR or immunohistochemistry. Unfortunately, the limited amount of genetic material available from cervical cytobrush samples precluded additional validation studies. The pilot study was also unable to assess the impact of co-infections, which were not differently distributed between groups but may contribute to the variability in host gene expression by acting as confounding factors. For example, we detected hookworm infection by microscopy in over 10% of women. Hookworms have been shown to exert immunomodulatory effects on the cervical mucosa [80], and even past helminth infections could alter gene expression, potentially creating similar gene profiles in women with and without *S. haematobium* infection. Moreover, epigenetic memory after infections can have long lasting effects on gene expression [81], which has been documented in peripheral blood with *S. mansoni* [82]. We also could not assess a causal relationship between *S. haematobium* infection, treatment, and gene expression changes due to lack of pre-infection samples. Together, these limitations could have led to the inability to identify pathways altered between those with and without *S. haematobium* infection at baseline, though differential expression of individual genes occurred. It is also possible that the exclusion of contaminated samples for RNA-Seq reduced statistical power and may have introduced bias. Reassuringly, we found that there were no sociodemographic or clinical differences between those whose samples were included and those that were contaminated. Further, due to limited number of samples available from women after treatment, RNA-Seq data was combined from all women with parasitological clearance post-praziquantel regardless of time point of sample collection and duration since treatment was given.

While these findings provide a clear signal of differences in gene expression between infected and uninfected women, as well as between women with parasitological clearance post-treatment, they do not establish

a definitive causal link. Instead, our results raise important hypotheses regarding how *S. haematobium* infection might influence the molecular environment in the female genital tract and how treatment might alter this landscape over time. Our findings underscore the need for additional studies to explore the long-term effects of treatment of schistosome infection on cervical health. The residual effects of *S. haematobium* eggs, treatment-induced inflammation, and chronic tissue damage could explain the persistent alterations in gene expression observed in women post-praziquantel treatment. Specifically, it will be critical to determine whether, in the weeks to months after successful parasitological clearance, altered gene expression in the cervical mucosa may even temporarily increase the risk for tumorigenesis, HPV acquisition or persistence, or both. This is particularly needed since the parasite's role in cervical cancer remains unclear [83–87]. Future studies should include larger cohorts with extended follow-up periods to better understand the trajectory of these gene expression changes and their clinical implications. Further, our work may lay the foundation for additional investigations into a genital tract biomarker that could diagnose FGS with high sensitivity and specificity. These findings could have important reproductive health implications for the estimated 40 million girls and women with *S. haematobium* infection worldwide [2].

## Supporting information

**S1 Fig.  Plots of two-dimensional projection of differential gene expression, by *S. haematobium* infection, cleared or persistently infected status.**
(TIF)

**S1 Table.  Lists of differentially expressed genes and top canonical pathways identified by IPA.**
(DOCX)

**S2 Table.  Lists of differentially expressed genes and top canonical pathways identified by IPA.**
(DOCX)

**S3 Table.  Lists of differentially expressed genes and top canonical pathways identified by IPA.**
(DOCX)

**S4 Table.  Lists of differentially expressed genes and top canonical pathways identified by IPA.**
(DOCX)

**S5 Table.  De-identified data set.**
(XLSX)

## Acknowledgments

We express our sincere gratitude to the study participants and to the study team for their excellent work. We thank the genomics core team at Weill Cornell for their invaluable assistance with our experiments and guidance. Additionally, we extend our thanks to Drs. Mary Charleson and Carol Mancuso who lead the Master's program Clinical Epidemiology & Health Services Research, and Dr. Kyu Rhee, who leads the Burroughs Wellcome Physician Scientist Program at Weill Cornell. Their ongoing support and insightful guidance have been indispensable for this study. Lastly, we thank Dr. Bishoy Faltas at Weill Cornell for reviewing our manuscript and for his insightful suggestions.

## Author contributions

**Conceptualization:** Anna M. Mertelsmann, Myung Hee Lee, Sheridan F Bowers, Jennifer A. Downs, John M. Changalucha.

**Data curation:** Anna M. Mertelsmann, Jane K Maganga, Myung Hee Lee, Maureen Ward, Adrian Y Tan, Sheridan F Bowers, Loyce Mhango, Jennifer A. Downs, John M. Changalucha.

**Formal analysis:** Anna M. Mertelsmann, Jane K Maganga, Myung Hee Lee, Adrian Y Tan, Jennifer A. Downs.

**Funding acquisition:** Anna M. Mertelsmann, Jane K Maganga, Jennifer A. Downs.

**Investigation:** Anna M. Mertelsmann, Jane K Maganga, Myung Hee Lee, Loyce Mhango, Saidi Kapiga, Kathryn M. Dupnik, Humphrey D. Mazigo, Jennifer A. Downs, John M. Changalucha.

**Methodology:** Anna M. Mertelsmann, Jane K Maganga, Myung Hee Lee, Maureen Ward, Adrian Y Tan, Sheridan F Bowers, Loyce Mhango, Danielle de Jong, Paul L.A.M. Corstjens, Govert J. van Dam, Kathryn M. Dupnik, Jennifer A. Downs, John M. Changalucha.

**Project administration:** Anna M. Mertelsmann, Jane K Maganga, Maureen Ward, Jennifer A. Downs, John M. Changalucha.

**Resources:** Anna M. Mertelsmann, Jennifer A. Downs.

**Software:** Anna M. Mertelsmann.

**Supervision:** Saidi Kapiga, Kathryn M. Dupnik, John M. Changalucha.

**Validation:** Anna M. Mertelsmann, Jane K Maganga, Myung Hee Lee, Adrian Y Tan, Danielle de Jong, Paul L.A.M. Corstjens, Govert J. van Dam, Kathryn M. Dupnik, Jennifer A. Downs, John M. Changalucha.

**Visualization:** Anna M. Mertelsmann.

**Writing – original draft:** Anna M. Mertelsmann, Jane K Maganga, Myung Hee Lee, Maureen Ward, Adrian Y Tan, Jennifer A. Downs.

**Writing – review & editing:** Anna M. Mertelsmann, Jane K Maganga, Myung Hee Lee, Maureen Ward, Adrian Y Tan, Sheridan F Bowers, Loyce Mhango, Danielle de Jong, Paul L.A.M. Corstjens, Govert J. van Dam, Saidi Kapiga, Kathryn M. Dupnik, Humphrey D. Mazigo, Jennifer A. Downs, John M. Changalucha.

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
