## [Decision Letter · Decision Letter 0]

15 Nov 2024

Response to Reviewers
Revised Manuscript with Track Changes
Manuscript

Shaden Kamhawi

co-Editor-in-Chief

Paul Brindley

co-Editor-in-Chief

**Additional Editor Comments (if provided):**

Thank you for submitting your manuscript to PLOS Neglected Tropical Diseases (PNTD-D-24-01412 - "Schistosoma haematobium Infection Promotes Oncogenic Gene Expression in Cervical Mucosa, with Enhanced Effects Following Treatment").

Expert reviewers in the fields of schistosomiasis and infectious diseases have evaluated your manuscript. They found that your study provides valuable insights into the impact of S. haematobium infection on oncogenic gene expression in cervical mucosa. However, they also identified areas for revision before the manuscript can be fully accepted by PNTD.

For example, reviewers suggested that you clearly address the limitations of the study, particularly the small number of participants enrolled and the high number of RNA-seq samples that failed to pass quality control. Additionally, they were concerned that the lack of significant pathways when comparing women with and without S. haematobium infection might be due to the data analysis pipeline used. They recommend reanalyzing the dataset with a modified pipeline to confirm the findings.

Finally, please ensure that all data used in this manuscript (including all phenotype data) are publicly available, with a functional link to the repository included in the manuscript.

The reviewers' comments are enclosed for your consideration, and I hope their feedback will be helpful for strengthening your analysis and manuscript overall. Please address all reviewers' comments before resubmission.

Again, thank you for your interest in PLOS Neglected Tropical Diseases.

Sincerely,

Winka Le Clec'h

**Journal Requirements:**
**Reviewers' Comments:**

**Key Review Criteria Required for Acceptance?**

**Methods**

-Are the objectives of the study clearly articulated with a clear testable hypothesis stated?

-Is the study design appropriate to address the stated objectives?

-Is the population clearly described and appropriate for the hypothesis being tested?

-Is the sample size sufficient to ensure adequate power to address the hypothesis being tested?

-Were correct statistical analysis used to support conclusions?

-Are there concerns about ethical or regulatory requirements being met?

Reviewer #1: (No Response)

Reviewer #2: It is unclear if the sample size was adequate, have mentioned this in the uploaded review. It is an exploratory paper and has value as such. Otherwise affirmative.

**Results**

-Does the analysis presented match the analysis plan?

-Are the results clearly and completely presented?

-Are the figures (Tables, Images) of sufficient quality for clarity?

Reviewer #1: (No Response)

Reviewer #2: The information is there but needs more succinct portrayal as written in the uploaded review.

**Conclusions**

-Are the conclusions supported by the data presented?

-Are the limitations of analysis clearly described?

-Do the authors discuss how these data can be helpful to advance our understanding of the topic under study?

-Is public health relevance addressed?

Reviewer #1: (No Response)

Reviewer #2: Conclusions are supported by the data. Limitations have been described but need to be concentrated as a section in the discussion. The authors discuss how these data can be helpful to advance our understanding of the topic and public health relevance is addressed but needs to be laid out properly. The discussion is the weakest section.

**Editorial and Data Presentation Modifications?**

Reviewer #1: Very minor typo:

Lines 91-92 “In S. haematobium-associated bladder cancer, S. haematobium eggs are classified as Group 1 carcinogen” should be “a Group 1 carcinogen”

Reviewer #2: (No Response)

**Summary and General Comments**

Reviewer #1: This is a very interesting manuscript but I do have methodological concerns:

24 samples failed to pass QC for RNA-Seq and only 39 passed. Likewise, many women (7) were not eligible for continued follow up after praziquantel treatment. The authors should better emphasize these important limitations

The lack of significant pathways (using IPA) comparing women with and without Sh is concerning. The authors should consider analyzing their data by modifying their pipeline (i.e., GSEA).

The genes involved in oncogenesis in Table 4 – what is the direction of differential gene expression for women who cleared infection post-praziquantel vs. women with baseline Sh?

Reviewer #2: The paper is interesting but it's a small sample size, making it more of a hypothesis generating paper. However, I do think it should be published and PNTD is a good forum, likely read by many who could make use of this information

PLOS authors have the option to publish the peer review history of their article (what does this mean? ). If published, this will include your full peer review and any attached files.

**Do you want your identity to be public for this peer review?** For information about this choice, including consent withdrawal, please see our Privacy Policy .

Reviewer #1: No

Reviewer #2: No

**Figure resubmission:****Reproducibility:** To enhance the reproducibility of your results, we recommend that authors of applicable studies deposit laboratory protocols in protocols.io, where a protocol can be assigned its own identifier (DOI) such that it can be cited independently in the future. Additionally, PLOS ONE offers an option to publish peer-reviewed clinical study protocols. Read more information on sharing protocols at https://plos.org/protocols?utm_medium=editorial-email&utm_source=authorletters&utm_campaign=protocols

---

## [Decision Letter · Decision Letter 1]

7 Jul 2025

Schistosoma haematobium Infection is Associated with Oncogenic Gene Expression in Cervical Mucosa, with Enhanced Effects Following Treatment

Dear Dr. Mertelsmann,

Thank you for submitting your manuscript to PLOS Neglected Tropical Diseases. After careful consideration, we feel that it has merit but does not fully meet PLOS Neglected Tropical Diseases's publication criteria as it currently stands. Therefore, we invite you to submit a revised version of the manuscript that addresses the points raised during the review process.

Please submit your revised manuscript within 60 days. If you will need more time than this to complete your revisions, please reply to this message or contact the journal office at plosntds@plos.org. Please include the following items when submitting your revised manuscript:

We look forward to receiving your revised manuscript.

Kind regards,

Winka Le Clec’h, PhD

Academic Editor

Krystyna Cwiklinski

Section Editor

Shaden Kamhawi

co-Editor-in-Chief

Paul Brindley

co-Editor-in-Chief

**Additional Editor Comments (if provided):**

Dear Dr. Mertelsmann,

Thank you for submitting your manuscript to PLOS Neglected Tropical Diseases (PNTD-D-24-01412, "Schistosoma haematobium Infection Promotes Oncogenic Gene Expression in Cervical Mucosa, with Enhanced Effects Following Treatment").

Expert reviewers in the fields of schistosomiasis and infectious diseases have evaluated your revised manuscript. They appreciated the revisions you have made. However, one reviewer has suggested additional areas for improvement before the manuscript can be fully accepted by PNTD.

In particular, validation of the nine genes identified as most differentially expressed in your RNA-seq analysis using RT-qPCR is essential to confirm the reliability of the observed expression patterns.

It is also important to discuss whether cervical cancer in this context may be related to somatic mutations in genes induced by the presence of *S. haematobium* , and/or whether it is associated with inflammation, fibrosis, and alterations to the epithelial cell microenvironment caused by the presence of *S. haematobium* eggs.

These are valid scientific questions that need to be addressed.

Finally, please ensure that all data used in this manuscript are publicly available, with a functional link to the data repository included in the manuscript. Sequence data should be submitted to the SRA repository.

The reviewers’ comments are enclosed for your consideration. I hope their feedback will be helpful in further strengthening your analysis and manuscript. Please address all reviewers' comments before resubmitting.

Again, thank you for your interest in PLOS Neglected Tropical Diseases.

Sincerely,

Winka Le Clec'h

Associate Editor

**Reviewers' Comments:**

Reviewer's Responses to Questions

**Key Review Criteria Required for Acceptance?**

**Methods**

-Are the objectives of the study clearly articulated with a clear testable hypothesis stated?

-Is the study design appropriate to address the stated objectives?

-Is the population clearly described and appropriate for the hypothesis being tested?

-Is the sample size sufficient to ensure adequate power to address the hypothesis being tested?

-Were correct statistical analysis used to support conclusions?

-Are there concerns about ethical or regulatory requirements being met?

Reviewer #3: Objectives are clear, the study design is apropriate, the population appropriate, the sample size is small, statistical analysis is correct, ethical and regulatory requirements seems to be well done.

Reviewer #4: (No Response)

**Results**

-Does the analysis presented match the analysis plan?

-Are the results clearly and completely presented?

-Are the figures (Tables, Images) of sufficient quality for clarity?

Reviewer #3: No problems here, also.

Reviewer #4: (No Response)

**Conclusions**

-Are the conclusions supported by the data presented?

-Are the limitations of analysis clearly described?

-Do the authors discuss how these data can be helpful to advance our understanding of the topic under study?

-Is public health relevance addressed?

Reviewer #3: Conclusions are not supported by data, and they need to be clear. Authors discussion is not helpful to advance our understanding of this topic.

Reviewer #4: (No Response)

**Editorial and Data Presentation Modifications?**

Reviewer #3: (No Response)

Reviewer #4: (No Response)

**Summary and General Comments**

Reviewer #3: Manuscript PNTD-D-24-01412R1 entitled “Schistosoma haematobium Infection is Associated with Oncogenic Gene Expression in Cervical Mucosa, with Enhanced Effects Following Treatment” authored by Anna M. Mertelsmann et al.

Global comments: Authors have choice an interesting, relevant and pertinent scientific topic. This is a strong work developed by scientists with domain of conceptual and instrumental tools, medical and scientific. They need to be stimulated to go on working on this scientific topic: urogenital schistosomiasis. However, results described in the manuscript does not sustain authors’ initial proposal. Small amount of patients? Associated infections? Controversial strategy? Authors studding cytobrush samples state “altered cervical mucosal gene expression with oncogenic potential in patients infected with S. haematobium”. What this means? Statistic relevance derived from a p value <0.05 for selected genes is not enough. The nine genes, selected, and expressed differentially should be validated with another test, for instance, Real Time PCR. Because selected patients are too complex with associated infections. “Women with S. haematobium infections compared to uninfected women, had gene expression consistent with enhanced oncogenesis in genital mucosa”. This is not clear to me and need to be proved, I am afraid. And it is not clear also the reason to select LINC02084 gene to discuss. Maybe because is thought to be related with other cancers discussions. Schistosoma haematobium is assumed as a carcinogenic pathogen. However, the mechanisms implicated in carcinogenesis are not yet clear. Cervical cancer related with S. haematobium infection is so important to be studied. The epistemology of this form of cancer is related with somatic mutations in genes or, on contrary, is related with inflammation, fibrosis and epithelial cells microenvironment alterations? In my point of view, somatic mutations are epiphenomena.

Reviewer #4: (No Response)

PLOS authors have the option to publish the peer review history of their article (what does this mean? ). If published, this will include your full peer review and any attached files.

**Do you want your identity to be public for this peer review?** For information about this choice, including consent withdrawal, please see our Privacy Policy .

Reviewer #3: **Yes: ** José M. Correia da Costa

Reviewer #4: No

**Figure resubmission:**

**Reproducibility:**



---

## [Decision Letter · Decision Letter 2]

12 Sep 2025

Dear Dr. Mertelsmann,

We are pleased to inform you that your manuscript 'Schistosoma haematobium Infection is Associated with Oncogenic Gene Expression in Cervical Mucosa, with Enhanced Effects Following Treatment: A pilot study' has been provisionally accepted for publication in PLOS Neglected Tropical Diseases.

Best regards,

Winka Le Clec’h, PhD

Academic Editor

Krystyna Cwiklinski

Section Editor

Shaden Kamhawi

co-Editor-in-Chief

Paul Brindley

co-Editor-in-Chief

p.p1 {margin: 0.0px 0.0px 0.0px 0.0px; line-height: 16.0px; font: 14.0px Arial; color: #323333; -webkit-text-stroke: #323333}span.s1 {font-kerning:Reviewer's Responses to Questions

**Key Review Criteria Required for Acceptance?**

**Methods**

-Are the objectives of the study clearly articulated with a clear testable hypothesis stated?

-Is the study design appropriate to address the stated objectives?

-Is the population clearly described and appropriate for the hypothesis being tested?

-Is the sample size sufficient to ensure adequate power to address the hypothesis being tested?

-Were correct statistical analysis used to support conclusions?

-Are there concerns about ethical or regulatory requirements being met?

Reviewer #3: -Are the objectives of the study clearly articulated with a clear testable hypothesis stated? Yes

-Is the study design appropriate to address the stated objectives? yes

-Is the population clearly described and appropriate for the hypothesis being tested? yes

-Is the sample size sufficient to ensure adequate power to address the hypothesis being tested? No

-Were correct statistical analysis used to support conclusions? yes

-Are there concerns about ethical or regulatory requirements being met? No

**Results**

-Does the analysis presented match the analysis plan?

-Are the results clearly and completely presented?

-Are the figures (Tables, Images) of sufficient quality for clarity?

Reviewer #3: -Does the analysis presented match the analysis plan? yes

-Are the results clearly and completely presented? yes

-Are the figures (Tables, Images) of sufficient quality for clarity? yes

**Conclusions**

-Are the conclusions supported by the data presented?

-Are the limitations of analysis clearly described?

-Do the authors discuss how these data can be helpful to advance our understanding of the topic under study?

-Is public health relevance addressed?

Reviewer #3: -Are the conclusions supported by the data presented? yes

-Are the limitations of analysis clearly described? yes

-Do the authors discuss how these data can be helpful to advance our understanding of the topic under study? yes

-Is public health relevance addressed? yes

**Editorial and Data Presentation Modifications?**

Reviewer #3: No

**Summary and General Comments**

Reviewer #3: I accept the publication of this manuscript according to its reviewed form; a project pilot with more work to be done. The novelty is cervical cancer.

PLOS authors have the option to publish the peer review history of their article (what does this mean? ). If published, this will include your full peer review and any attached files.

**Do you want your identity to be public for this peer review?** For information about this choice, including consent withdrawal, please see our Privacy Policy .

Reviewer #3: **Yes: ** José Manuel Correia da Costa

---

## [Editor Report · Acceptance letter]

Dear Dr. Mertelsmann,

We are delighted to inform you that your manuscript, " 

Schistosoma haematobium Infection is Associated with Oncogenic Gene Expression in Cervical Mucosa, with Enhanced Effects Following Treatment: A pilot study," has been formally accepted for publication in PLOS Neglected Tropical Diseases.

Best regards,

Shaden Kamhawi

co-Editor-in-Chief

Paul Brindley

co-Editor-in-Chief
